# The Role of Mitophagy in Glaucomatous Neurodegeneration

**DOI:** 10.3390/cells12151969

**Published:** 2023-07-30

**Authors:** Dimitrios Stavropoulos, Manjot K. Grewal, Bledi Petriti, Kai-Yin Chau, Christopher J. Hammond, David F. Garway-Heath, Gerassimos Lascaratos

**Affiliations:** 1Department of Ophthalmology, King’s College Hospital, London SE5 9RS, UK; stavropoulos.dim@gmail.com; 2Department of Ophthalmology, 417 Veterans Army Hospital (NIMTS), 11521 Athens, Greece; 3NIHR Biomedical Research Center, Moorfields Eye Hospital and UCL Institute of Ophthalmology, London EC1V 9EL, UK; 4Division of Optometry and Visual Science, School of Health Sciences, City, University of London, London EC1V 0HB, UK; 5Department of Clinical & Movement Neurosciences, UCL Queens Square Institute of Neurology, London NW3 2PF, UK; 6Section of Ophthalmology, School of Life Course Sciences, King’s College London, London SE1 7EH, UK; 7Department of Ophthalmology, St Thomas’ Hospital, London SE1 7EH, UK

**Keywords:** mitophagy, glaucoma, mitochondrial dysfunction, glaucomatous neurodegeneration, genetics, mitochondria, primary open-angle glaucoma

## Abstract

This review aims to provide a better understanding of the emerging role of mitophagy in glaucomatous neurodegeneration, which is the primary cause of irreversible blindness worldwide. Increasing evidence from genetic and other experimental studies suggests that mitophagy-related genes are implicated in the pathogenesis of glaucoma in various populations. The association between polymorphisms in these genes and increased risk of glaucoma is presented. Reduction in intraocular pressure (IOP) is currently the only modifiable risk factor for glaucoma, while clinical trials highlight the inadequacy of IOP-lowering therapeutic approaches to prevent sight loss in many glaucoma patients. Mitochondrial dysfunction is thought to increase the susceptibility of retinal ganglion cells (RGCs) to other risk factors and is implicated in glaucomatous degeneration. Mitophagy holds a vital role in mitochondrial quality control processes, and the current review explores the mitophagy-related pathways which may be linked to glaucoma and their therapeutic potential.

## 1. Introduction

Glaucoma is the primary cause of irreversible blindness worldwide [1], the second cause in the UK, and a leading cause of certifications for sight impairment [2]. The term glaucoma refers to a group of conditions characterized by progressive optic neuropathy, which occurs through slow and continuing apoptotic degeneration of retinal ganglion cells’ (RGCs) somata and axons, leading to a loss of visual function [3,4,5]. Our understanding of the pathophysiological mechanisms of glaucoma remains incomplete. Hypotheses, such as the vascular theory and the mechanical theory of glaucoma, were formed to explain how elevated intraocular pressure (IOP) harms the optic nerve (ON) [6,7]. Reduction in IOP is currently the only modifiable risk factor, established by several long-term clinical trials [8,9,10], to reduce the progression of glaucoma. However, patients deteriorate at all IOP levels and despite receiving IOP-lowering therapy [11], suggesting that other factors confer susceptibility. Furthermore, clinical studies emphasize the insufficient results of current IOP-lowering therapeutic approaches, with over 40% of glaucoma patients progressing after seven years in one study [12] and 60% showing statistically significant deterioration after an average of eight years of follow-up in another study [13]. As a result, it is widely accepted that other risk factors modulate the susceptibility of an eye to IOP. This has led to a considerable body of research on neurodegenerative mechanisms and the role of mitochondrial dysfunction in glaucoma [14]. Dysfunction of mitochondria has been strongly implicated in glaucomatous degeneration in multiple glaucoma models in recent years [15,16,17,18]. Mitochondrial dysfunction is capable of increasing the susceptibility of RGCs to stress from other risk factors, such as elevated intraocular pressure (IOP), light exposure and vascular insufficiency [19]. Mitophagy is a mitochondrial quality control process crucial in securing mitochondrial quantity and quality [20], and increasing evidence from various experimental studies indicates the importance of impaired mitophagy as a feature of glaucomatous degeneration [21,22,23]. This paper summarizes the evidence for the role of mitophagy-related genes in the pathogenesis of human glaucoma and in animal glaucoma models, and potential therapeutic avenues for glaucoma based on a modification of the mitophagy-related pathways.

## 2. Mitochondria in Glaucoma

The role of mitochondria in the pathogenesis of glaucoma has gained increasing interest as they are considered potential therapeutic targets [24,25]. Mitochondria are cytosolic organelles that hold a key role in generating energy in human cells in the form of adenosine triphosphate (ATP) through oxidative phosphorylation (OXPHOS) and other pathways [18]. Mitochondria also regulate calcium homeostasis in cells, intermediary metabolism, and the balance between reactive oxygen species (ROS) generation and scavenging [26]. The number of mitochondria per cell varies, depending on the energy demands of a cell. RGCs consume enormous amounts of energy to support axon potential generation in unmyelinated axons and to maintain axonal transport across the high-pressure gradient as axons leave the eye at the optic nerve head (ONH). The former is evidenced by the distribution and trafficking of mitochondria within RGC axons [27], and the latter is evidenced by the presence of giant mitochondria in ONH astrocytes, which provide support to RGC axonal metabolism at an anatomical site of particular vulnerability [28]. These huge energy demands make this part of the ON in the eye particularly susceptible to mitochondrial dysfunction [29]. ATP production and efficiency and number of mitochondria in RGCs decline with aging [30]. ROS are mainly produced through leakage of electrons from the respiratory chain and, under physiological conditions, ROS are detoxified by cellular antioxidants. When an impairment is present in this equilibrium, oxidative damage may induce apoptotic RGC death [31]. Studies have shown that although apoptosis is actively executed by caspases, RGC death may happen independently of caspase-mediated pathways [32]. These data suggest that both apoptosis and non-apoptotic cell death play a critical role in the death of RGCs. Wang et al. [33] suggest a non-apoptotic cell death mechanism of RGCs termed “paraptosis”, which may occur simultaneously with apoptosis and autophagy in glaucomatous RGCs and happens in two stages. Excessive production of ROS leads to mitochondrial impairment in glaucomatous RGCs, and mitochondrial damage may provoke RGC paraptosis.

Furthermore, nitric oxide, tumor necrosis factor alpha (TNFα), glutamate and other toxic substances released from glial cells in glaucoma are considered a possible path that leads to RGCs’ death due to a compromised energetic state when mitochondria are unable to maintain their expected function [34]. Glutamate excitotoxicity in glaucoma has been linked to increased calcium influx into mitochondria, causing the opening of the mitochondrial permeability transition pore (MPTP) of the inner mitochondrial membrane (IMM), increased ROS production, activation of apoptotic pathways and RGC degradation [35,36,37]. The human eye is also exposed every day to sunlight and artificial light, and it has been proposed that certain wavelengths of light can cause an enhanced generation of ROS in RGCs’ mitochondria and may lead to neuronal apoptosis [38,39].

In human glaucoma, mitochondrial dysfunction has been reported in peripheral blood mononuclear cells, tenon’s ocular fibroblasts, the trabecular meshwork and the lamina cribrosa. Vallabh et al. [40] demonstrated that basal respiration (oxygen consumption rate, OCR) was significantly reduced in glaucomatous tenon’s ocular fibroblasts compared to non-glaucomatous ones and demonstrated their impaired mitochondrial cellular bioenergetics. There is also evidence that susceptibility to IOP is related to mitochondrial function. Lascaratos et al. [41] demonstrated that patients with ocular hypertension who have not developed glaucoma despite experiencing raised IOP over several years have better lymphocyte mitochondrial function when compared to glaucoma patients with normal IOP (normal-tension glaucoma) and controls of a similar age. Additionally, Petriti et al. [42] showed that systemic mitochondrial function in the peripheral blood lymphocytes of patients with normal-tension glaucoma (NTG) was lower compared to patients with glaucoma and high IOP (high-tension glaucoma; HTG), suggesting that, in the absence of raised IOP as a clear risk factor for glaucoma, systemic mitochondrial dysfunction may play a critical role in the pathogenesis of glaucomatous neurodegeneration. Degeneration of RGCs’ mitochondria is an early feature of human glaucoma [43], and mitochondrial dysfunction has been identified in lamina cribrosa [44] and trabecular meshwork (TM) cells [45] of glaucomatous eyes. 

Studies with animal glaucoma models, which investigated the effects of elevated IOP, indicate a reduction in the number of healthy mitochondria with normal membrane potential, decreased mitochondrial cristae volume, smaller mitochondrial size and decreased mitochondrial movement within ONH axons after chronic IOP elevation [46,47,48,49]. Differential expression of genes involved in mitochondrial dysfunction and oxidative stress pathways is an early feature in a DBA/2J mouse glaucoma model, and age-dependent declines of nicotinamide adenine dinucleotide (NAD+) and glutathione in the retina render RGCs susceptible to damage from raised IOP [49]. 

The role of mitochondria in glaucoma susceptibility is further supported by genetic studies [50]. The three Mendelian genes, *myocilin*, *TANK-binding kinase 1 (TBK1)* and *optineurin (OPTN)*, have been linked to glaucoma, with the last two genes being mitophagy-related genes [51]. Khawaja et al. [52], using gene set analyses of a large cohort of patients, identified mitochondrial enzyme pathways critical to the pathogenesis of primary open-angle glaucoma (POAG). Moreover, polymorphisms in the *OPTN* gene (which mediates mitophagy) are associated with open-angle glaucoma (OAG) with normal lOP (NTG) and younger age of onset of HTG [53]. Inherited defects in mitochondria are associated with characteristic optic neuropathies, such as Leber’s hereditary optic neuropathy (mitochondrial genome mutations) and dominant optic atrophy (DOA) (mutations in a nuclear gene encoding inner mitochondrial membrane proteins) [54]. Interestingly, the clinical phenotype of these mitochondrial optic neuropathies may mimic NTG [55].

## 3. Mitophagy

Autophagy refers to a procedure of degradation and recycling of intracellular materials and inoperative organelles, which preserves cellular homeostasis [56]. It is subclassified into macroautophagy, microautophagy and chaperone-mediated autophagy. These three types of autophagy are completed through different mechanisms, but the final goal is the same: transportation of intracellular materials to lysosomes, where they can be degraded by hydrolytic enzymes [57]. In order to remain intact, mitochondria undergo several processes such as fusion (junction of two mitochondria into one), fission (separation of one mitochondrion into two), biogenesis (creation of new mitochondria) and mitophagy (degradation of mitochondria by autophagy) [58]. Mitophagy is a selective subtype of macroautophagy and a mitochondrial quality control process in which dysfunctional mitochondria are degraded in lysosomes [59] (Figure 1). Based on the targeting signals in impaired mitochondria that commence the process, it is divided into ubiquitin-dependent mitophagy, ubiquitin-independent or receptor-based mitophagy, lipid-based mitophagy and micromitophagy, with the first two categories being the most common [60]. An accumulation of mitochondrial DNA (mtDNA) mutations and diminished mitochondrial activity are thought to contribute to the normal aging process [61] and are associated with various systemic diseases, including heart failure, diabetes and cancer [62]. Mitophagy, as a quality control mechanism, is considered critical to the prevention of human disease by retaining normal cellular function, while aberrant mitophagy is implicated in the pathogenesis of multiple metabolic, cardiovascular, skeletal muscle and neurodegenerative diseases, including Alzheimer’s disease, amyotrophic lateral sclerosis, and Duchenne muscle dystrophy [63]. Several studies have demonstrated the importance of mitophagy in glaucomatous degeneration [21,22,23], as well as in other ocular diseases [64]. In the next section, we explore in more detail the different mitophagy-related genes (Table 1) and the growing scientific evidence that links mitophagy to glaucoma.

## 4. Mitophagy-Related Genes and Glaucoma Pathogenesis 

### 4.1. TBK1

The *TBK1* gene (*TANK-binding kinase 1*; MIM#604834), localized on 12q14.2, encodes the serine/threonine protein kinase TANK-binding kinase 1 [119], which controls the activation of genes in the nuclear factor kappa-light-chain-enhancer of activated B cell (NF-κB) pathway [120]. TBK1 holds a major role in cell proliferation, innate immune response, autophagy and apoptosis [121]. TBK1 is activated upon mitochondrial depolarization and regulates mitophagy via phosphorylation of major autophagy receptors, such as OPTN, calcium-binding and coiled-coil domain 2 (NDP52/CALCOCO2), sequestosome-1 (SQSTM1/p62) and tax1-binding protein 1 (TAX1BP1), enhancing their ubiquitin- and microtubule-associated protein 1A/1B-light chain 3 (LC3)-binding domains and, therefore, facilitating their autophagosomal engulfment [122]. *TBK1* gene duplications and triplications have been associated with approximately 1% of NTG cases, including Asian [65,66], Caucasian [67,68] and African American patients [69]. A single case of a patient with exfoliation glaucoma and *TBK1* gene duplication was reported by Fingert et al. [70]. Moreover, it is proven that *TBK1* is expressed in RGCs, which constitute a crucial site of NTG pathogenesis [69,123], while in induced pluripotent stem cell (iPSC)-derived retinal ganglion cells, an extra copy of *TBK1* has been found to activate the crucial autophagy protein LC3-II [124]. Transgenic *TBK1* mice show traits of NTG and significant progressive loss of RGCs [71].

### 4.2. OPA1

The *OPA1* gene (*optic atrophy-1*; MIM#605290; chr3q28-q29), which encodes a dynamin-related protein located in the inter-membrane space and is implicated in mitochondrial morphology [125], was identified by Delettre et al. [126]. OPA1 orchestrates fusion of the IMM [127] and ensures the maintenance of cristae integrity [128]. Studies have shown that mutations of *OPA1* disrupt IMM fusion, resulting in DOA [129,130]. In addition, Liao et al. [131] showed that apart from mitochondrial fragmentation, OPA1 loss leads to increased mitophagy and mitochondrial mislocalization. Moreover, Frezza et al. [132] proved, via electron microscopy, that OPA1 controls apoptosis by regulating cytochrome c redistribution and cristae remodeling. *OPA1* is expressed in the inner plexiform layer, the outer plexiform layer, the inner nuclear layer and the myelinated region beyond the lamina cribrosa within the ON [133,134,135]. Wang et al. [136] used three donated human eyeballs with no history of ocular disease and proposed that *OPA1* is expressed all the way from the prelaminar portion through the lamina cribrosa to the retrolaminar myelinated portion. Furthermore, Hu et al. [21] used purified RGCs from rats and chronic hypertensive glaucoma rats and proved that *OPA1* overexpression protects RGCs by intensifying fusion and enhancing parkin RBR E3 ubiquitin protein ligase (PARKIN)-mediated mitophagy. A study that investigated peripheral blood leucocytes from 43 POAG patients and 27 controls conducted by Bosley et al. [72] evidenced that OPA1 expression was significantly lower in the POAG patients compared to the controls. Aung et al. [73] found that polymorphisms in the *OPA1* gene were associated with NTG in two Caucasian patient cohorts. The single-nucleotide polymorphism (SNP) IVS8+4C/T was strongly associated with NTG in both cohorts, and the second SNP IVS8+32T/C was only associated with NTG occurrence in the first cohort. Another study by Aung et al. [137] could not demonstrate an association of these two SNPs with HTG. A study by Yu-Wai-Man et al. [74] in the North East England also demonstrated that the CT/TT compound genotype at VS8+4 and IVS8+32 is a strong genetic causal factor for NTG, but not HTG. Milanowski et al. [75] investigated *OPA1* polymorphisms in the Polish population and showed that the genotype CC and allele C of rs9851685 OPA1 polymorphism are NTG risk factors, whereas the TT genotype and T allele of this polymorphism are protective factors against NTG. Mabuchi et al. [138] investigated 194 Japanese patients with NTG and 191 with HTG and identified an association of the SNP IVS8+32T/C with NTG, but their study also revealed that this polymorphism is associated with earlier age of disease onset in HTG patients and should be considered a genetic risk factor for both. This result was not validated by the study conducted by Liu et al. [139], which demonstrated no association between the *OPA1* polymorphisms and a POAG phenotype that includes increased IOP in Caucasian, African American and Ghanaian populations. Several other studies also failed to demonstrate a correlation between *OPA1* and NTG in Greek [140], Korean [141], African Caribbean [142], Chinese [143], Mexican [144], German [76], Singaporean and Indian patient populations [145].

### 4.3. MFN1 and MFN2

*MFN1* (*mitofusin 1*; MIM#608506) and *MFN2* (*mitofusin 2*; MIM#608507) are genes that encode the transmembrane guanosine triphosphatases (GTPases) MFN1 and MFN2, respectively, which regulate fusion of the outer mitochondrial membrane (OMM) [146] where they are situated [147]. Chen et al. [148] demonstrated via linkage analysis that *MFN1* is localized on chromosome 3 at 3q25-26 and *MFN2* is localized on chromosome 1 at 1p36. MFN2 has a direct role in mitochondrial transport [149] and is proposed to regulate tethering and distance between mitochondria and the endoplasmic reticulum (ER) [150], modulating Ca^2+^ exchange between them [151]. A loss of mitofusins is associated with reduced autophagosome formation and fusion deficiencies of autophagosomes with lysosomes [152,153], which are a vital part of mitophagy. Mitofusins are ubiquitination targets by E3 ligases at the OMM [154]. MFN1 and MFN2 are polyubiquitinated [155] and degraded [156] following PARKIN activation. Mitochondrial fragmentation is necessary for the efficiency of mitophagy [157], and oxidative stress has been shown to cause mitochondrial fragmentation by regulating MFN2 [158]. Additionally, NTG has been associated in a German patient population with one SNP (rs2111534) in *MFN1* and three almost-consecutive SNPs (rs873458, rs2295281 and rs11588779) in *MFN2* [76]. Furthermore, an animal study with DBA and D2G mice conducted by Nivison et al. [159] demonstrated that MFN2 accumulates selectively in RGCs during glaucomatous degeneration and identified a phosphorylated MFN2 form that selectively accumulates in RGCs, but is absent in the ON. Finally, Milanowski et al. [75] investigated polymorphisms of *MFN1* and *MFN2* in the Polish population and showed that genotype GA of rs2111534 MFN1 polymorphism is a risk factor for HTG, while genotype AA of this polymorphism is protective against HTG, and genotype combinations of *OPA1* and *MFN2* are significantly associated with a higher or a lower risk of glaucoma.

### 4.4. AMBRA1

*AMBRA1* (*autophagy and beclin 1 regulator 1*; MIM#611359) is localized on chromosome 11 at 11p11.2, encoding a 1.300-amino-acid-long protein that acts as an initiator of phagophore formation in the nervous system, and its impairment leads to dysfunctional mitophagy and immoderate apoptosis [160]. Van Humbeeck et al. [161] showed that AMBRA1 interacts with PARKIN in adult mouse brain and under basal culture conditions, and PARKIN-AMBRA1 interaction increases during mitochondrial depolarization. In addition, Strappazzon et al. [162] demonstrated that AMBRA1 connects via its LC3-interacting region (LIR) motif to the autophagosome LC3 adapter after the induction of mitophagy, thereby enhancing PARKIN-mediated clearance and also regulating PARKIN-independent mitophagy. Furthermore, Di Rienzo et al. [163] proved, by using cell cultures, that AMBRA1 obstruction leads to increased apoptosis and decreased mitophagy induction, and upon mitochondrial depolarization, AMBRA1 is recruited to the OMM and connects to the ATPase family AAA domain-containing 3A (ATAD3A)-translocase of outer mitochondrial membrane 20 (TOMM20)-PTEN induced kinase 1 (PINK1) complex to enhance PINK1 aggregation. Old autophagy-impaired-AMBRA1 mice show higher RGC susceptibility [77], and heterozygous ablation of AMBRA1 leads to a remarkable reduction in RGC survivability after ischemia [78].

### 4.5. CAV1

*CAV1* (*caveolin-1*; MIM#601047) is mapped to chromosome 7q31.1 [164]. Nah et al. [165] showed that phosphorylated CAV1 triggers autophagy via interaction with beclin-1 (BECN-1) under oxidative stress. On the other hand, a study reported that CAV1 negatively modulates autophagy under deprivation circumstances by adjusting lysosomal function or interacting with autophagy-related 12-autophagy-related 5 (ATG12-ATG5) complex [166], and it binds in a phosphorylation-dependent manner to dynamin-related protein 1 (DRP1) and MFN2, negatively regulating mitophagy, fission and fusion and preventing PARKIN-mediated mitophagy [167]. A large genome-wide association study (GWAS) on POAG involving 1,236 cases and 34,877 controls in Iceland identified a common sequence variant at 7q31 and then replicated the association in 2,175 POAG cases and 2,064 controls, identifying a risk variant close to *CAV1* and *caveolin-2 (CAV2)*, which are expressed in the TM and RGCs [79]. Ozel et al. [80] conducted a GWAS and meta-analysis that identified *CAV1/CAV2* as a susceptibility locus for intraocular pressure. Moreover, 10 *CAV1/CAV2* SNPs were significantly associated with POAG, particularly among women [81], which was supported by another study in a Caucasian US population [82]. The polymorphism rs4236601 at the *CAV1-CAV2* locus was also associated with POAG and NTG in Chinese populations [83,84]. Association of rs4236601 and POAG was not detected in a Saudi cohort [168] and in a US cohort from Iowa [169].

### 4.6. OPTN

The *OPTN* gene (*optineurin*; MIM#602432) is localized on chromosome 10 at 10p13, encoding the protein optineurin which regulates vesicular transport [170] and NF-KB signaling [171]. Wong and Holzbaur [172] found, using live-cell microscopy, that optineurin is recruited to impaired mitochondria after PARKIN recruitment and stabilised via ubiquitin binding to ABIN and NEMO (UBAN) domain and eventually recruits LC3 via its LIR domain, leading to autophagosome formation. TBK1 phosphorylates OPTN’s UBAN domain, thereby enhancing its binding with the ubiquitin chains on mitochondria [122]. A study by Rezaie et al. [53] speculated that OPTN plays a neuroprotective role and identified *OPTN* mutations E50K and M98K as adult-onset glaucoma risk-associated factors. Studies using E50K transgenic animal models demonstrated that these animals experienced autophagosome–lysosome fusion impairment, age-dependent energy deficiencies, protein synthesis malfunction [85], visual impairment in contrast sensitivity [86] and retinal pathological phenotypes [87]. Furthermore, studies have shown an association of *OPTN* with glaucoma in Japanese [88], Indian [89,90], Chinese [91,92,93], Korean [94] and Finnish [95] patient populations. On the contrary, studies in Faroese [173], Spanish [174] and Taiwanese [175] glaucoma patients showed no association, and the last study also demonstrated a protective role of the variant c.-233+25C>G against glaucoma. Lastly, NTG patients with the *OPTN* E50K mutation were found to have a more severe glaucomatous phenotype with advanced optic disc cupping and smaller neuroretinal rim area at diagnosis, when compared to NTG patients without the mutation [96].

### 4.7. HK2

The *HK2* gene (*hexokinase 2*; MIM#601125) is localized on chromosome 2 at 2p12, encoding HK2, a glycolytic enzyme that boosts tumor glycolysis, progression and metastasis [176]. A study by McCoy et al. [177] identified HK2 as a modifier of PARKIN recruitment to mitochondria and demonstrated the crucial role of hexokinase activity for PARKIN recruitment and subsequent mitophagy. HK2 assists in the formation of a PINK1 complex that activates the PARKIN ubiquitin ligase to promote mitochondrial ubiquitylation and recruitment of ubiquitin-binding mitophagy receptors, such as OPTN [178]. A two-stage case-control study by Shi et al. [97] that tested 669 SNPs from the region of chromosome 2 in Japanese patients showed that *HK2* had significant association with POAG and NTG. Lastly, a study that investigated the relationship of the *HK2* gene and NTG in a Korean cohort showed that the *HK2* gene polymorphism rs678350 may contribute to NTG genetic susceptibility [98]. 

### 4.8. PARL

*PARL* gene (*presenilin-associated rhomboid-like* gene; MIM#607858), located on chromosome 3 at 3q27.1, encodes a mitochondrial protease that regulates the function of OPA1 [179] and influences apoptosis by controlling cytochrome c release via OPA1-dependent cristae remodeling [180]. PARL mediates cleavage of PINK1, and PARL knockdown is thought to lead to an amassing of ectopic *PINK1* expression at the mitochondrial surface [181], PARKIN recruitment and, subsequently, mitophagy activation [182]. Two neighboring SNPs in *PARL* (rs1000002 and rs1402003) were found to be associated with NTG in a German population, which was confirmed by using multimarker haplotype-based association testing [76]. In a Chinese cohort of 422 primary angle-closure glaucoma (PACG) patients and 400 control subjects, one SNP (rs3749446) in PARL was found to be associated with PACG, but following a conditional analysis, it did not remain significantly associated, suggesting that the association of this SNP was not independent [99].

### 4.9. TP53

The *TP53* gene (*tumor protein p53*; MIM#191170), mapped to chromosome 17 at 17p13.1 [183], encodes the transcription factor p53, whose activation promotes cell cycle arrest to permit DNA repair and/or apoptosis in order to prevent the propagation of cells with significant DNA damage [184]. P53 also regulates mitophagy through the PINK1-PARKIN pathway by inhibiting the mitochondrial translocation and activation of PARKIN, which diminishes mitophagy and leads to failure of cells to effectively remove damaged mitochondria [185]. Two studies involving Caucasian patient populations showed significant association between POAG and two SNPs (rs1042522 and rs17878362) in *TP53* [100,101]. The polymorphism rs1042522 was found to be associated with POAG in two Chinese cohorts, with the C allele frequency being higher in the POAG patients than in the controls in one study [102] and the G allele being more frequent in the other [143]. Blanco-Marchite et al. [103] showed that the *T*P53 p.R72P polymorphism acts as a glaucoma risk factor in Spanish patients and suggested a genetic interplay between *TP53* and *WD repeat domain 36 (WDR36)* variants in POAG susceptibility. Furthermore, *TP53* was found to be associated with POAG in a cohort of 65 unrelated POAG patients in Iran [104] and with PACG in a North Indian patient cohort [105]. In a meta-analysis with subgroup analyses controlling for ethnicity, the association between the *TP53* codon 72 polymorphism and POAG risk was detected in Asian populations, but not in Caucasian populations [106]. Another study involving Chinese POAG patients suggested the rs4938723 SNP in *TP53* may act as a protective factor against POAG [107], while Wiggs et al. [186] proposed that the P53 codon 72 PRO/PRO genotype is associated with early paracentral visual field defects in POAG patients. Nevertheless, various studies did not demonstrate significant association between rs1042522 and POAG in Polish [187], Turkish [188], Japanese [189], Australian [190], Brazilian [191] and Indian patient populations [192].

### 4.10. ROCK1

*ROCK1* gene (*Rho-associated coiled-coil containing protein kinase 1*; MIM#601702) is localized on chromosome 18 at 18q11.1 and encodes a homonymous protein that regulates myosin light chain 2 (MLC2) phosphorylation and peripheral actomyosin contraction and is consequently involved in actin cytoskeleton remodeling [193]. A recent study showed that ROCK activates several substrates including phosphatase and tensin homolog (PTEN), which is a negative regulator of PARKIN [194], and identified ROCK as a negative PARKIN-mediated mitophagy regulator [195]. *ROCK1* is expressed in the TM as demonstrated by a study using TM tissue derived from monkeys [196]. A study with 363 POAG patients and 213 healthy controls in Korea showed that the SNPs rs288979, rs1006881, rs35996865, rs10083915 and rs11873284 in *ROCK1* are associated with an increased risk of HTG but not with NTG [108]. The *ROCK1* polymorphism rs35996865 was found to have no association with POAG in a Turkish patient population [197].

### 4.11. TNFα

*TNF*α gene (*tumor necrosis factor alpha*; MIM#191160) is localized in chromosome 6 at 6p21.3, encoding a multifunctional proinflammatory cytokine that belongs to the tumor necrosis factor (TNF) superfamily [198]. *TNF*α overexpression increases autophagy and premature senescence in malignant melanoma cells due to mitochondrial dysfunction [199]. Moreover, quantitative proteomics demonstrates that TNFα activation of macrophages induces down-regulation of mitochondrial proteins through activation of mitophagy [200]. Interestingly, TNFα upregulation has been implicated in RGC apoptosis in human glaucomatous ONH [109] and in animal glaucoma models [110,111]. *TNF*α was also found to be associated with POAG in a study with 60 Chinese POAG patients and 103 controls, which demonstrated that the A allele (−308G>A) frequency of the promoter SNP rs1800629 was higher in the POAG patients compared to the controls [201]. Another study with a Chinese patient population showed that the G allele frequency of the promoter SNP rs1800629 was higher in HTG patients than in control subjects, while one TNF haplotype consisting of rs1799724 and rs1800629 was also significantly associated with HTG [143]. A study with a Caucasian patient population failed to prove the association between the two promoter SNPs (−308G>A and −238G>A) of *TNF*α and POAG [202], and a Japanese study revealed no significant association between three promoter SNPs (−308G>A, −857C>T and −863C>A) of *TNF*α and POAG, despite a potential interaction between *OPTN* and *TNF*α [203].

### 4.12. PARKIN

The *PARKIN* gene (*Parkin RBR E3 ubiquitin protein ligase*; MIM#602544) is localized on chromosome 6 at 6q26, encoding PARKIN, a member of RBR E3 ubiquitin ligases, which ubiquitinates outer mitochondrial membrane proteins upon the depolarization of mitochondria [204]. PARKIN orchestrates the autophagic clearance of impaired mitochondria [205]. PINK1 aggregation in depolarized mitochondria leads to PINK1-mediated phosphorylation of PARKIN and ubiquitin (UB) conjugates on mitochondrial substrates in close proximity, which ultimately triggers and recruits PARKIN to these mitochondria [206]. Overexpression of *PARKIN* exerts protective effects against age-related function loss in tissues and confers protection against senescence in various in vitro and in vivo models [112]. Additionally, *PARKIN* overexpression in a human neuroblastoma SH-SY5Y cell line was found to preserve mtDNA against oxidative damage and promote mtDNA repair [207]. *PARKIN* overexpression also repairs the mitophagy process and reduces mitochondrial abnormalities in Alzheimer’s disease [208], and decreases cell death in human mesenchymal stromal cells exposed to neurotoxicant 6-hydroxydopamine (6-OHDA) [209]. Overexpression of *PARKIN* has also been shown to protect retinal ganglion cells in experimental glaucoma [19]. Aging is a major risk factor for most neurodegenerative diseases, including glaucoma, and PARKIN is thought to hold a neuroprotective role in various models of neurodegeneration. Therefore, if we thoroughly investigate the mechanisms through which PARKIN exerts positive action on age-related neurodegenerative diseases, new therapeutic avenues may be revealed. 

### 4.13. DRP1

The *DRP1* gene (*dynamin-related protein 1*; MIM#603850), also known as *DNM1L (dynamin 1-like)*, is localized on chromosome 12 at 12p11.21, encoding DRP1 GTPase which facilitates mitochondrial fission [210]. Phosphorylation or dephosphorylation of serine sites S616 (phosphorylation) and S637 (dephosphorylation) regulates DRP1 activity [211]. When recruited, DRP1 creates a ring-like formation around the OMM, which undergoes GTP hydrolysis, leading to membrane compression and consequently scission [212]. DRP1 recruitment promotes the entry of Zn^2+^ into the mitochondrial matrix, the mitochondrial membrane potential is decreased, and, upon fission, the fragmented healthy mitochondria manage to restore their membrane potential and join the fission–fusion cycle again, whereas mitochondria that cannot accomplish this are eliminated via mitophagy [213]. Obstruction of DRP1-mediated mitochondrial fission prevents PARKIN-mediated mitophagy in HeLa cells and mouse embryonic fibroblasts (MEFs) [156]. DRP1 also facilitates mitophagy activation during hypoxia in HeLa cells [214]. Downregulation of DRP1 inhibits mitophagy, suppresses autophagosome formation, and leads to cell death at baseline and under stress conditions, such as ischemia/reperfusion in cardiomyocytes [215]. A study by Kim et al. [113] showed that oxidative stress activates mitochondrial fission and loss of RGCs in a mouse glaucoma model through an increase in DRP1 activity, and DRP1 inhibition may protect RGCs by maintaining mitochondrial integrity. Glaucomatous human ONH astrocytes with healthy N-Methyl-D-aspartic acid (NMDA) receptors show upregulation of DRP1 and its phosphorylation at serine 616 [114]. Lastly, a study by Edwards et al. [115] showed an increase in total DRP1 levels and a reduction in DRP1 phosphorylation at serine 637 (Ser637) in the retina in a mouse glaucoma model, leading to mitochondrial fragmentation and loss, as well as mitophagosome formation in RGCs.

### 4.14. UCP2

The *UCP2* gene (*uncoupling protein 2*; MIM#601693), is localized in chromosome 11 at 11q13.4, encoding the mitochondrial uncoupling protein 2, which belongs to the family of mitochondrial anion carrier proteins and uncouples oxygen consumption from ATP synthesis, while also minimizing ROS emission from the electron transport chain [216]. Forte et al. [217] showed that PARKIN is upregulated in cells overexpressing *UCP2* and suggested that UCP2 modulates mitophagy, since *UCP2* knockdown leads to mitophagy reduction and, consequently, to the accumulation of defective mitochondria. *UCP2* expression is altered during different stages of glaucoma and increases with elevated IOP, and overexpression of *UCP2* in RGCs significantly reduces cell death in mice with high IOP [116]. Moreover, a study that investigated the TM of patients with neovascular glaucoma who underwent trabeculectomy demonstrated that UCP2 was significantly decreased in the TM cells of neovascular glaucoma patients compared to the control group [117].

### 4.15. KEAP1 and NRF2

The *KEAP1* gene (*Kelch-like ECH-associated protein 1*; MIM#606016) is mapped to chromosome 19 at 19p13.2, encoding the homonymous protein, and the *NRF2* gene (*NF-E2-related factor 2*; #MIM600492) is localized in chromosome 2 at 2q31.2, encoding the transcription factor NRF2. Both KEAP1 and NRF2 are major regulators of redox homeostasis [218]. KEAP1 binds NRF2 through its C-terminal Kelch domain [219]. Two KEAP1 molecules and one NRF2 molecule form a trimer, and KEAP1 regulates the activity, stability and accumulation of NRF2 [220]. The KEAP-NRF2 stress response pathway is the main inducible defense against oxidative stress insults [221]. Zeb et al. [222] showed that KEAP1 is involved in the regulation of the PINK1-PARKIN pathway by ROS and ROS-induced mitophagy, since overexpression of *KEAP1* blocks PARKIN translocation from the cytosol to mitochondria and *KEAP1* silencing enhances mitophagy. Murata et al. [223] showed that NRF2 upregulates the *PINK1* gene (Figure 2) as a response to oxidative stress conditions. Lastly, a study by Naguib et al. [118] suggested that oxidative stress is an early event in the pathogenesis of glaucoma and revealed an antioxidant response mediated by the NRF2-KEAP1 pathway, which is activated by NRF2 phosphorylation.

## 5. Mitophagy as a Therapeutic Target

Since impaired mitophagy is implicated in glaucomatous degeneration, as described previously, modulating mitophagy may prove to be an important strategy to treat glaucoma. Reduction in IOP and higher aqueous humor outflow through the TM are achieved by ROCK inhibitors, as demonstrated in animal models [224]. In a discovery pipeline study using thousands of compounds to identify small molecules that increase PARKIN recruitment to damaged mitochondria and ensuing mitophagy, ROCK inhibitors (such as SR3677) were found to trigger mitophagy through increased *HK2* activation in flies exposed to a parkinsonian toxin that induces mitochondrial damage [195]. Apart from increasing *HK2* activity to promote mitophagy, ROCK inhibitors may hold a neuroprotective role by inhibiting the release of inflammatory cytokines in microglial cells [225]. In addition, ROCK inhibitors promote RGC regeneration and survival as demonstrated in various optic nerve injury models when delivered intravitreally [226,227], but they also reduce RGC loss and promote axonal regeneration when administered topically [228]. In a randomized phase II clinical trial against latanoprost, the ROCK inhibitor Netarsudil 0.02% demonstrated significant reductions of 5.7 mmHg in IOP after four weeks of dosing [229]. The FDA approved Netarsudil for clinical use in 2017 [230]. 

In a rat glaucoma model, adeno-associated virus 2-PARKIN (AAV2-PARKIN), which was used to overexpress PARKIN, led to a significant decrease in RGC loss and partial restoration of mitophagy levels under elevated IOP conditions, suggesting that clearance of impaired mitochondria through mitophagy is directly linked to better RGC survival [22]. Fucoxanthin, a carotenoid of the xanthophyll family with antioxidant and anti-inflammatory properties, also exerts protective effects on RGCs by enhancing PARKIN-mediated mitophagy under glutamate excitotoxicity [231]. Interestingly, a chronic ocular hypertension rat glaucoma model showed that during short-term IOP elevations, fucoxanthin decreases *PARKIN* expression and reduces the number of mitophagosomes and autophagosomes to prevent excessive damage caused by unrestrained mitophagy, while during long-term IOP elevations, fucoxanthin protects RGCs, improves mitochondrial health, increases *PARKIN* expression and enhances the PARKIN-mediated mitophagy pathway [232].A recent study by Zhuang et al. [233] showed that the small natural molecule S3 derived from diterpenoids could protect RGCs by enhancing PARKIN-mediated mitophagy under NMDA-induced excitotoxicity. Metformin, an FDA-approved mammalian target of rapamycin (mTOR) inhibitor, stimulates mitophagy by restoring PARKIN-mediated mechanisms and mTOR-dependent autophagy initiation [234], is associated with a reduced risk of OAG development in people with diabetes [235], and has been correlated with a reduced risk of OAG that could not be explained by IOP [236]. 

Hu et al. [21] injected hypertensive glaucoma model rats with the adeno-associated virus 2-OPA1 (AAV2-OPA1) and reported healthier mitochondria in the optic nerve RGCs, bigger mitochondrial surface area and enhanced mitochondrial fusion. Overexpression of *OPA1* increased *PARKIN* expression, induced PARKIN-mediated mitophagy and protected RGCs. Treatment with memantine, an NMDA glutamate receptor antagonist, promoted RGC survival in pre-glaucomatous mice by enhancing *OPA1* expression, reducing *DNM1* (a mouse homologue of *DRP1*) expression, and inhibiting OPA and cytochrome c release from mitochondria [237]. Two randomized double-masked, placebo-controlled, multicenter phase 3 clinical studies that lasted 48 months showed no benefit of orally administered memantine [238]. 

Moreover, an animal study by Hass and Barnstable [23] showed that deletion of *mitochondrial uncoupling protein 2 (UCP2)*, despite uncoupling the electron transport chain from ATP synthase activity and increasing ROS production, promotes mitophagy in cortical astrocyte cultures and retinal tissue, and ultimately decreases RGC loss, in male and female mice. This demonstrates a potential neuroprotective role and pharmacological compounds that inhibit *UCP2* may provide a novel therapeutic approach.

P62-mediated mitophagy inducer (PMI) is another promising neuroprotective agent that inhibits *KEAP1* and activates *NF-E2-related factor 2 (NRF2)*, leading to the expression of antioxidant genes, including *sequestosome 1 (p62/SQSTM1)*, and enhanced mitophagy [239]. DRP1 may also serve as an auspicious target for a potential antibody-based therapy for glaucoma, since intravitreal injection of an anti-DRP1 antibody in an animal glaucoma model protected RGCs, decreased apoptosis as shown using western blot analysis, and improved retinal functionality as observed using electroretinography [240]. 

Coenzyme Q10 (CoQ10) is an essential component for the transport of electrons in mitochondria, and electron microscopy studies on fibroblasts from patients with CoQ10 deficiency showed increased lysosomal markers and mitochondrial degradation via mitophagy [241]. Using a retinal ischemia/reperfusion animal model, Nucci et al. [242] demonstrated that intraocular CoQ10 administration reduces glutamate increase and affords neuroprotection. A retrospective study by Alpogan et al. [243] evaluated the impact of CoQ10 eye drops combined with vitamin E on the retinal nerve fiber layer (RNFL) and ganglion cell complex (GCC) thickness in patients with OAG and showed a statistically significant beneficial effect of CoQ10 on the RNFL. Moreover, Parisi et al. [244] showed an improvement in inner retinal function using pattern electroretinogram and in visual evoked potential amplitudes in patients with OAG who were treated with CoQ10 and vitamin E eye drops over 12 months.

Pharmacological regulation of mitophagy beyond ophthalmology attracts great interest, as mitophagy holds a significant role in highly prevalent chronic diseases [63]. Bhansali et al. [245,246] showed that metformin promotes mitophagy by enhancing important mitophagy genes, such as *PINK1*, and preserves mitochondrial health in mononuclear cells of type 2 diabetes patients. Ursolic and oleanolic acids have also been suggested to have anti-tumorigenic effects that are dependent on their mitophagy-enhancing properties [247]. Finally, gerontoxanthone I and macluraxanthone have been shown to regulate mitophagy through the PINK1-PARKIN pathway and are thought to exert protective effects against myocardial ischemia-reperfusion injury [248].

## 6. Discussion/Conclusions

It is widely accepted that glaucoma is likely caused by the interactions between multiple genes and environmental factors [249]. Current IOP-lowering therapeutic approaches, as indicated by clinical studies, fail to adequately control glaucoma progression in the long term in a significant number of patients. Emerging evidence from genetic and experimental studies reveals an increasing number of genes involved in mitophagy that may contribute to glaucoma susceptibility. Mitophagy-related processes and pathways may act as targets for novel gene-based glaucoma therapies and pharmacological interventions. Understanding the mechanisms that confer resistance to glaucoma progression would be critical to enable the development of a genetic or biochemical test that may identify glaucoma patients with mitophagy defects who are at risk of progression and/or more likely to respond to treatment, thus promoting the field of personalized medicine in glaucoma. It is important to note that apart from single genes and single-gene variants, gene–gene and gene–environment interactions should be further investigated in pursuance of more thorough knowledge on the subject. This review presents the increasing body of evidence that mitochondrial dysfunction and mitophagy-related genetic pathways may contribute to glaucoma pathogenesis and therapeutics.

## Figures and Tables

**Figure 1 cells-12-01969-f001:**
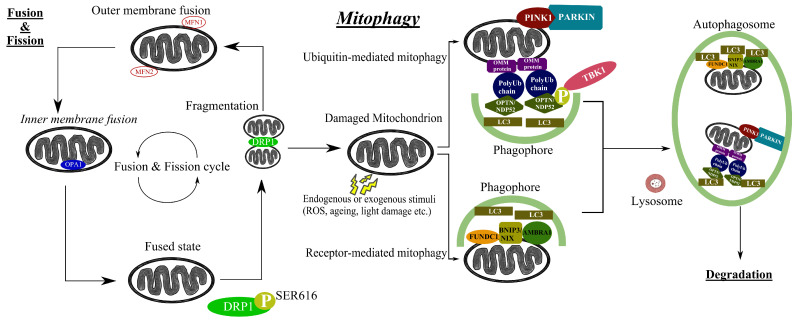
Summary of mitochondrial quality control process and mitophagy pathways. Mitochondrial dynamics comprise constant cycles of fusion and fission. When a mitochondrion is depolarized or dysfunctional, it is subjected to elimination. Impaired mitochondria are removed from a cell through two main mitophagy pathways, the PTEN-induced kinase 1 (PINK1)/ Parkin RBR E3 ubiquitin protein ligase (PARKIN) pathway, which is a ubiquitin-mediated pathway, and the receptor-mediated pathway. At the beginning of the PINK1/PARKIN pathway, PINK1 is stabilized in the outer mitochondrial membrane (OMM) and recruits PARKIN, which is an E3 ubiquitin ligase. Consequently, PARKIN polyubiquitinates various OMM proteins. In turn, the polyubiquitin chains are identified by adaptor proteins such as optineurin (OPTN), which interact with microtubule-associated protein 1A/1B-light chain 3 (LC3), and as a result, the impaired mitochondrion gets isolated via a phagophore in the double-membraned autophagosome. During the receptor-mediated pathway, mitophagy receptors, such as BCL2-interacting protein 3 (BNIP3)/NIX or autophagy and beclin 1 regulator 1 (AMBRA1), attach to the interaction region of LC3, allowing them to interact directly with the phagophore. In the aftermath, the impaired mitochondrion will end up in the autophagosome where, after fusion with lysosomes, it will get degraded.

**Figure 2 cells-12-01969-f002:**
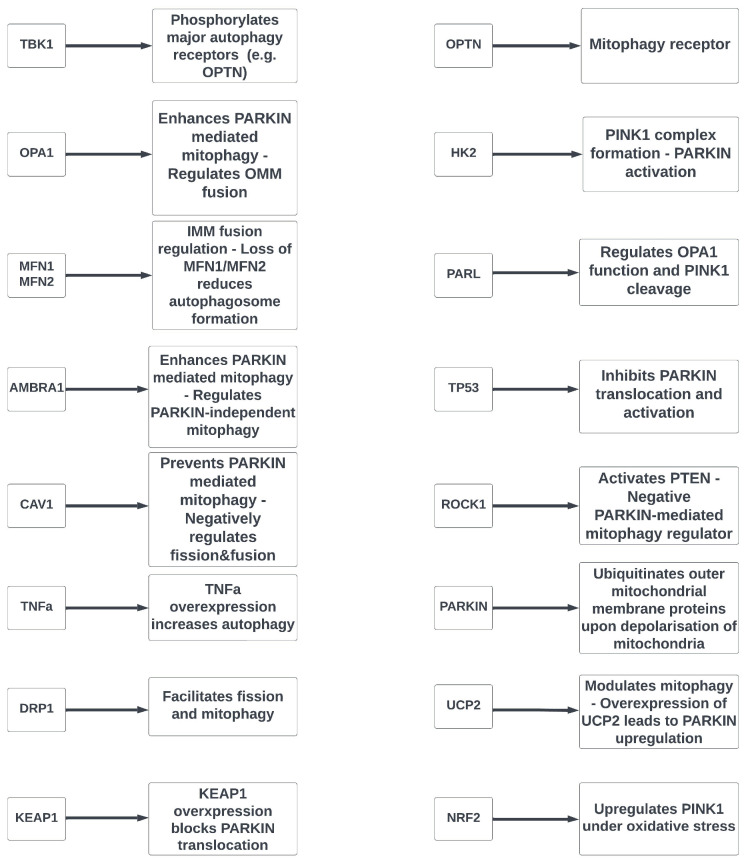
Synopsis of the main effects that proteins derived from mitophagy-related genes have on mitophagy and mitophagy pathways.

**Table 1 cells-12-01969-t001:** Mitophagy-related genes linked to glaucoma.

Gene Symbol	Gene Name	MIM ID	Chromosomal Location	References
*TBK1*	*TANK-binding kinase 1*	604834	12q14.2	[65,66,67,68,69,70,71]
*OPA1*	*Optic atrophy 1*	605290	3q28-q29	[72,73,74,75]
*MFN1*	*Mitofusin 1*	608506	3q25-q26	[75]
*MFN2*	*Mitofusin 2*	608507	1p36.2	[76]
*AMBRA1*	*Autophagy and beclin 1 regulator 1*	611359	11p11.2	[77,78]
*CAV1*	*Caveolin-1*	601047	7q31.1	[79,80,81,82,83,84]
*OPTN*	*Optineurin*	602432	10p13	[85,86,87,88,89,90,91,92,93,94,95,96]
*HK2*	*Hexokinase 2*	601125	2p12	[97,98]
*PARL*	*Presenilin-associated rhomboid-like*	607858	3q27.1	[99]
*TP53*	*Tumor protein p53*	191170	17p13.1	[100,101,102,103,104,105,106,107]
*ROCK1*	*Rho-associated coiled-coil containing protein kinase 1*	601702	18q11.1	[108]
*TNFα*	*Tumor necrosis factor alpha*	191160	6p21.3	[109,110,111]
*PARKIN*	*Parkin RBR E3 ubiquitin protein ligase*	602544	6q26	[112]
*DRP1*	*Dynamin-related protein 1*	603850	12p11.21	[113,114,115]
*UCP2*	*Uncoupling protein 2*	601693	11q13.4	[116,117]
*KEAP1*	*Kelch-like ECH-associated protein 1*	606016	19p13.2	[118]
*NRF2*	*NF-E2-related factor 2*	600492	2q31.2	[118]

## Data Availability

Not applicable.

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
