# Peer review of "The Role of Mitophagy in Glaucomatous Neurodegeneration"

_cells, 2023, doi:10.3390/cells12151969_

Round 1

Reviewer 1 Report

This is just a lengthy review of the literature on a topic that, right now, has a very limited interest for scientifically based hypotheses on the etio- pathogenesis of the glaucomas. No new personal contribution of the authors.

Sufficient.

Author Response

Response to Reviewer 1

Comment 1: This is just a lengthy review of the literature on a topic that, right now, has a very limited interest for scientifically based hypotheses on the etio- pathogenesis of the glaucomas. No new personal contribution of the authors.

Response: Thank you for your feedback. We have made every effort to provide a comprehensive structured review of the emerging role of mitophagy in glaucoma. Mitochondrial dysfunction is increasingly considered an important risk factor for glaucoma and mitophagy is critical for mitochondrial quality control. Clinical trials are currently under way in several continents to explore the role of mitochondria-enhancing treatments in delaying glaucoma progression. In this review we have tried to summarise the evidence that links mitophagy to glaucoma and highlight the potential therapeutic role of mitophagy-related pathways.

Reviewer 2 Report

The manuscript entitled, “The role of mitophagy in glaucomatous neurodegeneration” presents the growing evidence that mitochondrial dysfunction and mitophagy-related genetic pathways may contribute to glaucoma pathogenesis and therapeutics. The data presented in this review is very interesting and it is well written, however there are some minor comments that the authors should address:

·         Line 93: the authors included "skin fibroblasts"" and I assume they mean “tenon’s ocular fibroblasts”

·         Lines 101 to 108: The work published by Petriti et al. did not include a control group in their study. Please specify where the exposed results were obtained with respect to the control group or delete this part of the paragraph.

·         Include a small diagram of the pathways by which each of the proteins derived from each mitophagy-related gene acts to make it easier for the reader to understand the review.

·         There are many abbreviations throughout the manuscript, please include a list of abbreviations in the manuscript to make it easier for the reader to follow the review.

·         In point 5 of the manuscript "Mitophagy as a therapeutic target" there are several genes and proteins, such as UCP2 and KEAP1 and NRF2, that are not discussed in the main body of the paper, and it is odd that they are discussed only in this point if they are possible main therapeutic targets. Include an item in the main body of the manuscript for each of these genes or one that includes other genes related to mitophagy and glaucoma that picks up these genes and proteins.

·         Restructure item 5 grouping treatments by therapeutic targets: PARKIN, ROCK, DRP1.....

Author Response

Response to Reviewer 2

Comment 1: [Line 93: the authors included "skin fibroblasts"" and I assume they mean “tenon’s ocular fibroblasts”]

Response: Thank you very much. Changed to “tenon’s ocular fibroblasts”.

Comment 2: Lines 101 to 108: The work published by Petriti et al. did not include a control group in their study. Please specify where the exposed results were obtained with respect to the control group or delete this part of the paragraph.

Response: Thank you and we have corrected accordingly.

“Petriti et al. [41] showed that systemic mitochondrial function in peripheral blood lymphocytes from patients with normal tension glaucoma (NTG) was lower compared to patients with glaucoma and high IOP (high tension glaucoma; HTG) suggesting that, in the absence of raised IOP as a clear risk factor for glaucoma, systemic mitochondrial dysfunction may be playing a critical role in the pathogenesis of glaucomatous neurodegeneration.”

Comment 3: Include a small diagram of the pathways by which each of the proteins derived from each mitophagy-related gene acts to make it easier for the reader to understand the review.

Response: Thank you very much for your suggestion. A diagram was created and added in line 504 which depicts all the proteins derived from each mitophagy-related gene and their main action in relation to mitophagy pathways.

Comment 4: There are many abbreviations throughout the manuscript, please include a list of abbreviations in the manuscript to make it easier for the reader to follow the review.

Response: Thank you. An abbreviations list has been added at the end of the text including all the abbreviations in the manuscript in alphabetical order (lines 607-674).

Comment 5: In point 5 of the manuscript "Mitophagy as a therapeutic target" there are several genes and proteins, such as UCP2 and KEAP1 and NRF2, that are not discussed in the main body of the paper, and it is odd that they are discussed only in this point if they are possible main therapeutic targets. Include an item in the main body of the manuscript for each of these genes or one that includes other genes related to mitophagy and glaucoma that picks up these genes and proteins.

Response: Thank you so much for your suggestion. Two new paragraphs were created in the main body of the manuscript for the genes UCP2, KEAP1 and NRF2, as requested (lines 472 – 503).

Comment 6: Restructure item 5 grouping treatments by therapeutic targets: PARKIN, ROCK, DRP1....

Response: Thank you very much for your comment. We have revised item 5 accordingly.

Reviewer 3 Report

Stavropoulos et al. discuss the role of mitophagy-related genes in the pathogenesis of glaucoma and potential therapeutic approaches based on modifying mitophagy pathways. Glaucoma is a leading cause of irreversible blindness worldwide and is characterized by progressive optic neuropathy and loss of visual function. The current understanding of the pathophysiological mechanisms is incomplete and the role of mitophagy has been discussed in many research papers recently. This review paper is helpful to summarize and contextualize these original research papers. The manuscript highlights that mitochondrial dysfunction has been strongly implicated in glaucomatous degeneration, increasing the susceptibility of retinal ganglion cells (RGCs) to stress. Mitophagy plays a crucial role in maintaining mitochondrial function. Impaired mitophagy has been identified as an aspect of glaucomatous degeneration. The review manuscript highlights evidence for the involvement of mitophagy-related genes in glaucoma models. It discusses genes such as TBK1OPA1MFN1 & MFN2AMBRA1CAV1OPTNHK2PARLTP53ROCKTNFαPARKIN and DRP1. The authors also consider the potential therapeutic target of mitophagy-related genes, such as PARKIN and OPA1, in glaucoma treatment. I noticed a few minor points for revision, as discussed below.

1)    In the first line of the introduction, a reference is needed for the statement that glaucoma is the primary cause of irreversible blindness. 

2)    Furthermore, the authors only mention apoptotic cell death of RGCs in glaucomatous retinas in this manuscript (e.g. lines 33-36, 80-81), while not mentioning other cell death mechanisms that have been described in glaucoma and can be associated with mitophagy. For the sake of completeness, these should at least be mentioned.

3)    Figure 1 illustrates the described processes clearly and comprehensibly, but seems blurry. The text fonts and sizes should be adjusted uniformly. The symbols used in the figure are inconsistent, and the overall appearance of the diagram appears hand-drawn and provisional. The figure should be revised and improved to ensure a more polished visual representation. Additionally, the protein names should be placed directly next to their corresponding symbols instead of being mentioned in the legend, because the molecules of interest are depicted at a very small scale. These modifications would make the figure easier to read and understand. 

4)    Table 1 should be embedded in the text and “(Table 1)” deleted from the heading “4. Mitophagy-related genes and glaucoma pathogenesis”. Also, the last paragraph is not formatted in block style.

5)    The review “Mitochondrial dysfunction in glaucomatous degeneration” by Zhang et al. 2023 (PMID: 37206187) discusses the role of mitochondria in healthy and glaucomatous retina comprehensively. The authors may consider referencing this article. 

The manuscript is well-written and logically structured. 

Author Response

Response to Reviewer 3

Comment 1: In the first line of the introduction, a reference is needed for the statement that glaucoma is the primary cause of irreversible blindness. 

Response: Thank you very much for the suggestion. A reference has been added in the first line.

Comment 2: Furthermore, the authors only mention apoptotic cell death of RGCs in glaucomatous retinas in this manuscript (e.g. lines 33-36, 80-81), while not mentioning other cell death mechanisms that have been described in glaucoma and can be associated with mitophagy. For the sake of completeness, these should at least be mentioned.

Response: Thank you very much for your comment. We have revised accordingly. We added a part in lines 81-88, where other mechanisms are mentioned.

Comment 3: Figure 1 illustrates the described processes clearly and comprehensibly, but seems blurry. The text fonts and sizes should be adjusted uniformly. The symbols used in the figure are inconsistent, and the overall appearance of the diagram appears hand-drawn and provisional. The figure should be revised and improved to ensure a more polished visual representation. Additionally, the protein names should be placed directly next to their corresponding symbols instead of being mentioned in the legend, because the molecules of interest are depicted at a very small scale. These modifications would make the figure easier to read and understand.

Response: Thank you for your helpful suggestion. Figure 1 has been revised accordingly. We have upscaled its quality, edited the text fonts and sizes and revised the symbols to make them consistent and easier to read. We removed all the blurry parts of the figure and protein names were placed directly on their corresponding symbols, as requested.

Comment 4: Table 1 should be embedded in the text and “(Table 1)” deleted from the heading “4. Mitophagy-related genes and glaucoma pathogenesis”. Also, the last paragraph is not formatted in block style.

Response: Thank you for your comment. Table 1 has been embedded in the text, “(Table 1)” has been deleted from the heading “4. Mitophagy-related genes and glaucoma pathogenesis” and the last paragraph has been formatted in block style.

Comment 5: The review “Mitochondrial dysfunction in glaucomatous degeneration” by Zhang et al. 2023 (PMID: 37206187) discusses the role of mitochondria in healthy and glaucomatous retina comprehensively. The authors may consider referencing this article. 

Response: Thank you very much for your suggestion. The review by Zhang et al. is a very interesting paper and the reference has been added in the manuscript.

Round 2

Reviewer 1 Report

Sorry to confirm my previous judgement on this submission.

Sorry to confirm my previous judgement on this submission.

Author Response

Thank you again for your comments.